# Preparation, Structure, and Electrical Properties of Cobalt-Modified Bi(Sc_3/4_In_1/4_)O_3_–PbTiO_3_–Pb(Mg_1/3_Nb_2/3_)O_3_ High-Temperature Piezoelectric Ceramics

**DOI:** 10.3390/mi12121556

**Published:** 2021-12-13

**Authors:** Zhijiang Chen, Na Lin, Zhao Yang, Juan Zhang, Kefei Shi, Xinhao Sun, Bo Gao, Tianlong Zhao

**Affiliations:** 1Institute of Materials, China Academy of Engineering Physics, Mianyang 621907, China; chenzhijiang@caep.cn (Z.C.); linna@caep.cn (N.L.); yangchao@hrbeu.edu.cn (Z.Y.); 2School of Microelectronics, Xidian University, Xi’an 710071, China; zhangjuan@stu.xidian.edu.cn (J.Z.); 21111213184@stu.xidian.edu.cn (K.S.); sunxinhao2018@gmail.com (X.S.)

**Keywords:** BSI–PT–PMN ceramic, Co ion doping, high temperature piezoelectric application

## Abstract

Cobalt-modified 0.40Bi(Sc_3/4_In_1/4_)O_3_–0.58PbTiO_3_–0.02Pb(Mg_1/3_Nb_2/3_)O_3_ ceramics (abbreviated as BSI–PT–PMN–*x*Co) were produced by conventional two-step solid-state processing. The phase structure, micro structure morphology, and electrical properties of BSI–PT–PMN–*x*Co were systematically studied. The introduction of Co ions exerted a significant influence on the structure and electrical properties. The experiment results demonstrated that Co ions entered the *B*-sites of the lattice, resulting in slight lattice distortion and a smaller lattice constant. The average grain size increased from ~1.94 μm to ~2.68 μm with the increasing Co content. The optimized comprehensive electrical properties were obtained with proper Co-modified content 0.2 wt.%. The Curie temperature (*T*_c_) was 412 °C, the piezoelectric constant (*d*_33_) was 370 pC/N, the remnant polarization (*P*_r_) was 29.2 μC/cm^2^, the relatively dielectric constant (*ε*_r_) was 1450, the planar electromechanical coupling coefficient (*k*_p_) was 46.5, and the dielectric loss (tan*δ*) was 0.051. Together with the enhanced DC resistivity of 10^9^ Ω cm under 300 °C and good thermal stability, BSI–PT–PMN–0.2Co ceramic is a promising candidate material for high-temperature piezoelectric applications.

## 1. Introduction

Lead zirconate titanate (PZT) systems represent most of the market share of piezoelectric materials because of their excellent piezoelectric performance, simple preparation process, and low cost [1,2,3]. However, their relatively low Curie temperature (*T*_c_), which means their piezoelectric properties decrease quickly with rising environmental temperatures, restrict their high-temperature application in petrochemical, aerospace, and other industries [4,5,6].

In 2001, Eitel et al. [7] first reported a new high-temperature piezoelectric material based on (1−*x*)Bi(Me)O_3_–*x*PbTiO_3,_ where Me^3+^ refers to Sc^3+^, Y^3+^, In^3+^, Yb^3+^, etc. Preferably, (1−*x*)BiScO_3_–*x*PbTiO_3_ (BS–PT) ceramics exhibit a morphotropic phase boundary (MPB) at *x* = 0.64, with a high Curie temperature of 450 °C and excellent piezoelectric performance (*d*_33_ = 460 pC/N), comparable to commercial soft PZT ceramics [8]. Later, extensive studies, such as the single-element doping [9,10,11,12,13,14,15], composition modification [16,17,18,19,20,21,22], and the introduction of a third component [23,24,25,26,27,28,29,30], focused on improving the piezoelectric, dielectric or mechanical properties of BS–PT ceramics were reported. For example, single-element doping refers to the substitution of Sc^3+^ by other +3 valence cations, such as In^3+^, Co^3+^, et al. Unfortunately, although the *T*_c_ is still over 400 °C, the piezoelectric properties decrease significantly. For example, for the In^3+^-modified, BS–PT based solid solutions Bi(Sc_3/4_In_1/4_)O_3_–PbTiO_3_ (BSI–PT), the *T*_c_ reaches 457 °C, whereas the *d*_33_ reduces to only 201 pC/N [16] for the MPB composition. The composition modification refers to the substitution of Sc^3+^ by composition ions such as (Mg_1/2_Ti_1/2_)^3+^, (Ni_1/2_Ti_1/2_)^3+^, (Zn_1/2_Zr_1/2_)^3+^, (Ni_1/2_Zr_1/2_)^3+^, etc. With proper composition ion modification, obtained improved piezoelectric properties can be obtained, together with considerable *T*_c_. For example, BS–PT–0.025Bi(Ni_1/2_Zr_1/2_)O_3_ solid solutions exhibit *d*_33_ of 480 pC/N and *T*_c_ 439 °C for MPB composition [20]. Another way to modify the electrical properties of the BS–PT binary system is to introduce a third component to form BS–PT–ABO_3_ ternary solid solutions. The electrical properties can be regulated due to the different chemical composition of ABO_3_. For example, the piezoelectric properties (*d*_33_ = 555 pC/N, *k*_p_ = 59%) and Curie temperature, of 408 °C, were obtained for the MPB composition in the BS–PT–Pb(Sn_1/3_Nb_2/3_)O_3_ system [27]. Furthermore, the BS–PT–LiNbO_3_ system possesses an enhanced *d*_33_ of 551 pC/N and reduced *T*_c_ of 337 °C in the vicinity of the MPB [29]. The BS–PT–BiGaO_3_ system exhibits an enhanced *T*_c_ of 511 °C and a reduced *d*_33_ of 102 pC/N [30].

In our previous study, we combined the first method and the third method to form a new ternary system Bi(Sc_3/4_In_1/4_)O_3_–PbTiO_3_–Pb(Mg_1/3_Nb_2/3_)O_3_ to modify the electrical properties of the BS–PT solid solution [31]. The optimal properties were obtained for the (0.98-*x*)BSI–*x*PT–0.02PMN ceramics at the MPB *x* = 0.58 of piezoelectric constant (*d*_33_) 403 pC/N, planar electromechanical coupling factor (*k*_p_) 47.2%, and remnant polarization (*P*_r_) 36.4 μC/cm^2^. The Curie temperature (*T*_c_) remained at 421°C, making the system much more suitable for high-temperature piezoelectric applications.

Furthermore, high electrical resistance is so necessary for high-temperature application that a large electric field can be applied during poling without breakdown or excessive charge leakage. As a common doping element in various piezoelectric material systems, cobalt is often used to improve the electrical properties of piezoelectric materials [32,33,34,35,36,37]. Co ion is commonly used as acceptor-type dopant to replace the *B*-site ion in perovskite materials. Generally, in the case of *B*-site substitution, the improvement of piezoelectric properties is not as good as with *A*-site substitution, but it can improve the high temperature resistivity greatly while maintaining the Curie temperature. In this study, we attempt to select a Co ion substitution for 0.40Bi(Sc_3/4_In_1/4_)O_3_–0.58PbTiO_3_–0.02Pb(Mg_1/3_Nb_2/3_)O_3_ ternary ceramics, aiming to generate good electrical performance while still providing a relatively high Curie temperature.

## 2. Experimental Procedure

The 0.40Bi(Sc_3/4_In_1/4_)O_3_–0.58PbTiO_3_–0.02Pb(Mg_1/3_Nb_2/3_)O_3_–*x*Co (*x* = 0, 0.2, 0.4, 0.6, 0.8 wt.%) ceramics were produced through conventional two-step solid state processing. The reagent-grade materials of Bi_2_O_3_ (99.8%), Sc_2_O_3_ (99.99%), In_2_O_3_ (99.99%), PbO (99%), TiO_2_ (99.5%), MgO (99%), Nb_2_O_5_ (99.9%), and Co_2_O_3_ (99.9%) were utilized as the raw materials and weighted by the stoichiometric amount. First, the powder MgO and Nb_2_O_5_ were mixed and calcined at 1100 °C for 4 h to synthesize the columbite precursor MgNb_2_O_6_. Next, the precursor MgNb_2_O_6_ and other raw materials were mixed together for 12 h by ball milling with alcohol. The powders were then dried, ground, granulated by milling with polyvinyl alcohol (PVA) as the binder, and then pressed into tablets with a diameter of 12.5 mm and a thickness of 1.5 mm under 120 MPa of pressure. Before sintering, a cold isostatic pressing process of 200 MPa was carried out on the tablets for 20 min. After burning out PVA at 650 °C, the tablets were sintered at 1120 °C for 3 h in a muffle furnace.

The X-ray diffraction (XRD) patterns of the BSI–PT–PMN samples were obtained by using an X-ray diffractometer (Cu Kα, λ = 1.5406, D/Max 2500, Rigaku, Tokushima, Japan). The fresh fracture surface microstructure of the samples was observed by using scanning electron microscopy (SEM) (S–4800, Hitachi, Tokyo, Japan). After polishing and ultrasonic cleaning, the ceramic samples were evenly coated with silver slurry and then fired at 600 °C. After cooling to room temperature, the ceramic samples were poled at 120 °C in a silicone oil bath under a DC electric field of 4.5 kV/mm for 20 min. The dielectric behavior as a function of temperature was analyzed using an impedance analyzer (4294A, Agilent, Palo Alto, California, USA). The piezoelectric coefficient *d*_33_ was measured using a quasi-static type meter (ZJ–3A, Institute of Acoustics, Beijing, China). The *P*-*E* hysteresis loops of the samples were characterized using a ferroelectric analyzer (TF1000, aixACCT, Aachen, Germany). The temperature dependence of the DC resistivity was measured by a Digit Multimeter (34410A 6½, Agilent, Palo Alto, CA, USA). When it was necessary to measure the thermal stability, the samples were heated to the set temperature and held for an hour. The piezoelectric coefficient *d*_33_ was measured when the samples cooled down to room temperature.

## 3. Results and Discussion

Figure 1 presents the X-ray diffraction patterns of BSI–PT–PMN–*x*Co (*x* = 0, 0.2 0.4, 0.6, 0.8 wt.%). From the figure, one can clearly observe that all the Co-doped ceramic samples presented the perovskite phase with a secondary phase of Co_2_O_3_ (Index PDF#02-0770), indicating that the Co modification did not change the crystal structure of BSI–PT–PMN ceramics significantly. Furthermore the characteristic peak moved slightly, to a high angle, as the Co content increased, which can be seen in Figure 1b. This phenomenon indicates the decrease in the lattice constant in this system. According to the Bragg equation (2*d*sinθ = *nλ*), the lattice constant was calculated and listed in Table 1. It can be seen that the lattice parameter *a* changed slightly and the lattice parameter *c* decreased with the increasing Co content *x*. It has been reported that Co ions exist in the valence state as Co^2+^ if the sintered temperature exceeds 1080 °C [38]. On account of the fact that the samples were sintered at 1120 °C in this study, the Co^3+^ and Co^2+^ were coexistent for Co_2_O_3_ used as dopants in this work. Furthermore, the radii of Co^3+^ and Co^2+^ were 0.63 Å and 0.72 Å, respectively, smaller than the radius of Ti^4+^ of 0.86 Å; Co^3+^ or Co^2+^ may have entered the lattice to replace Ti^4+^ in the *B*-sites for the BSI–PT–PMN ceramics and resulted in the decrease in the lattice constant. In addition, the *c*/*a* ratio decreased from 1.022 to 1.017 with the increasing Co content *x*, which exerted a significant effect on the electric properties.

Figure 2 presents the microstructure of the faces for BSI–PT–PMN–*x*Co (*x* = 0, 0.2, 0.4, 0.6, 0.8 wt %) sintered at 1120 °C. All the samples exhibited a dense structure with no obvious porosity, in accordance with the high relative density listed in Table 1. The average grain sizes of the BSI–PT–PMN–*x*Co samples were statistically analyzed and listed in in Table 1 using *Nano Measurer* software (V1.2, developed by Visual Basic 6.0 (Microsoft, Redmond, WA, USA), free from the Internet), with the number of the grains over 50. It was found that the average grain size increased from ~1.94 μm to ~2.68 μm as the Co content increased, suggesting that the doping of Co ions can reduce the sintering temperature and facilitate grain growth, which has been reported in other Co-doped piezoelectric ceramics [39].

Figure 3 depicts the curves of the temperature dependence of the dielectric permittivity (*ε*) for the BSI–PT–PMN–*x*Co ceramics. It can be seen that the dielectric peak of the BSI–PT–PMN–*x*Co ceramics became suppressed and broader when the measured frequency increased, and there was no frequency dispersion in any of the samples. The spectral line of the dielectric permittivity featured only one peak, which corresponded to the Curie temperature.

The Curie temperature ranged from 421 °C to 398 °C with increasing Co ion-doped content, which can be attributed to the enlarged tolerance factor ‘*t*’ caused by the *B*-site substitution. For the perovskite ferroelectric structure, the tolerance factor ‘*t*’ can be described as
$$t=\frac{r_{A}+r_{O}}{\sqrt{2}\left(r_{B}+r_{O}\right)}$$
where *r*_A_, *r*_B_, and *r**_O_* refer to the ionic radii of the *A*-site, *B*-site, and oxygen atom, respectively. As is already known, for the perovskite ferroelectric structure, there is a consistent relationship between the Curie temperature and the tolerance factor ‘*t*’: the lower tolerance factor ‘*t*’, the higher the Curie temperature. As discussed above, the introduced Co^3+^ and Co^2+^ replaced Ti^4+^ and entered into *B*-sites for BSI–PT–PMN ceramics. As a result, the tolerance factor ‘*t*’ enlarged and the Curie temperature declined due to the reduced *B*-sites effective ionic radius. Furthermore, typical dielectric diffuse phenomena can be found in this system, which can be described by the modified Curie–Weiss law as
$$\frac{1}{\varepsilon }-\frac{1}{\varepsilon_{m}}=\frac{{\left(T-T_{m}\right)}^{\gamma }}{C}$$
where *ε*_m_ is the maximum value of the dielectric permittivity *ε*, *T*_m_ is the temperature of *ε*_m_, *C* is the Curie–Weiss constant, and *γ* is the diffuseness coefficient value between 1 to 2, respectively. The values of *γ* of all the BSI–PT–PMN–*x*Co ceramics were calculated and presented in Figure 3. In this system, the dielectric diffuseness may be mainly attributed to the polar nanodomains generated by the relaxor PMN third component (0.02Pb(Mg_1/3_Nb_2/3_)O_3_ ), due to the slight change in the values of *γ* in line with the increasing Co content.

Figure 4 presents the *P*–*E* hysteresis loop behavior for the BSI–PT–PMN–*x*Co (*x* = 0, 0.2, 0.4, 0.6, 0.8) ceramics at room temperature under 1 Hz. The samples featured better saturated *P*–*E* loops when the Co ion-doped content increased. For the undoped BSI–PT–PMN ceramic, *P*_r_ was 36.4 μC/cm^2^ and *E*_c_ was 24.7 kV/cm. By doping the Co ions, *P*_r_ decreased and reached a minimum value of 25.1 μC/cm^2^, while *E*_c_ increased and reached a maximum value of 32.3 kV/cm at *x* = 0.8. This phenomenon may have been due to the Co ions entering the lattice to replace the Ti ions in the *B*-sites, resulting in oxygen vacancy, which plays a role in domain wall pinning.

Figure 5 presents the strain (*S*) versus the electric field (*E*) loop by applying an electric field of 40 kV/cm and 1 Hz of BSI–PT–PMN–*x*Co ceramics. The strain under a unipolar electric field of 40 kV/cm was 0.222, 0.205, 0.198, 0.188, and 0.182, with 0, 0.2%, 0.4%, 0.6% and 0.8% Co ion doping, respectively.

Furthermore, the relevant large-signal *d*_33_* and strain hysteresis *h* were calculated and listed in Table 2, where the strain hysteresis *h* can be described as
$$h=\frac{\Delta S}{S_{\max}}\times 100\%$$
where Δ*S* is the strain difference at half the maximal driven electric field in a cycle and the *S*_max_ is the strain maximum. It was found the large-signal *d*_33_* were over 450 pm/V for all these BSI–PT–PMN–*x*Co ceramics and the strain curve exhibited a good linearity with small hysteresis *h*, which is advantageous for piezoelectric actuator applications.

The electrical properties of the Co ion-doped BSI–PT–PMN ceramics at room temperature are listed in Table 2. It can be seen that the Curie temperature (*T*_c_), relatively dielectric constant (*ε*_r_), piezoelectric constant (*d*_33_), remnant polarization (*P*_r_), planar electromechanical coupling coefficient (*k*_p_), and dielectric loss (tan*δ*) monotonically decreased, while the coercive electric field (*E*_c_) monotonically increased, when the Co ion content increased from 0 to 0.8 wt.%.

Figure 6 presents the Arrhenius-type plots of conductivity as a function of temperature for BSI–PT–PMN with undoped and 0.2% doped Co ion. The activation energy *E*_ac_ was calculated according to the Arrhenius law: *σ*
*= σ*_0_exp(-*E*_ac_/*kT*), where *σ*, *E*_ac_, *k*, and *T* are conductivity, activation energy, Boltzmann constant, and temperature, respectively. The activation energy was 0.3 eV at 25 < *T* < 220 °C and 1.5 eV at 220 < *T* < 400 °C for the undoped BSI–PT–PMN, while the activation energy was 0.33 eV at 25 < *T* < 220 °C and 1.43 eV at 220 < *T* < 400 °C for 0.2% Co-doped BSI–PT–PMN. The value of 0.2% Co ion-doped BSI–PT–PMN ceramic was found to be slightly larger than the value of the undoped BSI–PT–PMN ceramic at high temperature.

It has been reported that for the BS–PT system, the activation energy *E*_ac_ for ionic conduction is to be ≥1 eV and for electronic conduction around 0.1 eV [40,41,42]. This indicates that ionic conduction plays a more important role than electronic conduction when T > 220 °C. In this study, Co ions were introduced into the BSI–PT–PMN system. The employment of Co ions may reduce the Pb leakage loss during calcination and sintering by trapping the bonded electrons and increase the resistivity. As a result, the 0.2% Co ion-doped BSI–PT–PMN ceramic exhibited a lower activation energy at 220 < *T* < 400 °C.

In addition, as can be seen in the inset of Figure 6, it was observed that the 0.2% Co ion-doped BSI–PT–PMN ceramic exhibited a higher resistivity value than the undoped BSI–PT–PMN ceramic (10^9^ Ω magnitude to10^8^ Ω magnitude) at 300 °C, which is advantageous for high-temperature actuator applications.

Figure 7 depicts the piezoelectric constant *d*_33_ as a function of temperature for BSI–PT–PMN–*x*Co (*x* = 0, 0.2, 0.4, 0.6, 0.8). All the piezoelectric properties were measured at room temperature after annealing the poled specimens at different temperatures for 1 h. All the samples displayed the same trend and the piezoelectric constant *d*_33_ remained almost unchanged from room temperature to 250 °C; it began to drop when the annealing temperature exceeded 250 °C, dropped quickly when the annealing temperature exceeded 300 °C, and finally dropped to zero near the Curie temperature point. For the ceramic with 0.2 wt.% Co ion doping, the value of the piezoelectric constant *d*_33_ at 325 °C could still reach 80% of room temperature. The results reveal that BSI–PT–PMN–0.2Co ceramics possesses better thermal stability than ceramics without Co ion doping. The harder domain switching [43] may explain this phenomenon.

## 4. Conclusions

High-temperature piezoelectric ceramic 0.40Bi(Sc_3/4_In_1/4_)O_3_–0.58PbTiO_3_–0.02Pb(Mg_1/3_Nb_2/3_)O_3_ (abbreviated as BSI–PT–PMN) doped with different amounts of Co ions (*x* = 0, 0.2, 0.4, 0.6, 0.8 wt.%) were prepared. The influences of Co ion doping on the phase transition, micro structure morphology, and electrical properties of the prepared BSI–PT–PMN–*x*Co ceramics were studied. It was found that Co ions entered the *B*-sites of the lattice, resulting in slight lattice distortion and a smaller lattice constant. With the increase in the doping of the Co ions, the grain size became larger; Co ions can promote the grain growth. The high temperature resistivity and temperature stability of the BSI–PT–PMN ceramics can be improved through an appropriate amount of Co ion doping. When the Co ion-doped content was 0.2 wt.%, the BSI–PT–PMN exhibited optimized comprehensive electrical properties: the Curie temperature (*T*_c_) was 412 °C, the piezoelectric constant (*d*_33_) was 370 pC/N, the remnant polarization (*P*_r_) was 29.2 μC/cm^2^, the relatively dielectric constant (*ε*_r_) was 1450, the planar electromechanical coupling coefficient (*k*_p_) was 46.5, and the dielectric loss (tan*δ*) was 0.051. BSI–PT–PMN–0.2Co ceramics feature the merits of relatively high Curie temperature, high resistivity at high temperature, large strain, good strain linearity, and good thermal stability. BSI–PT–PMN–0.2Co ceramics are suitable for high-temperature actuator applications.

## Figures and Tables

**Figure 1 micromachines-12-01556-f001:**
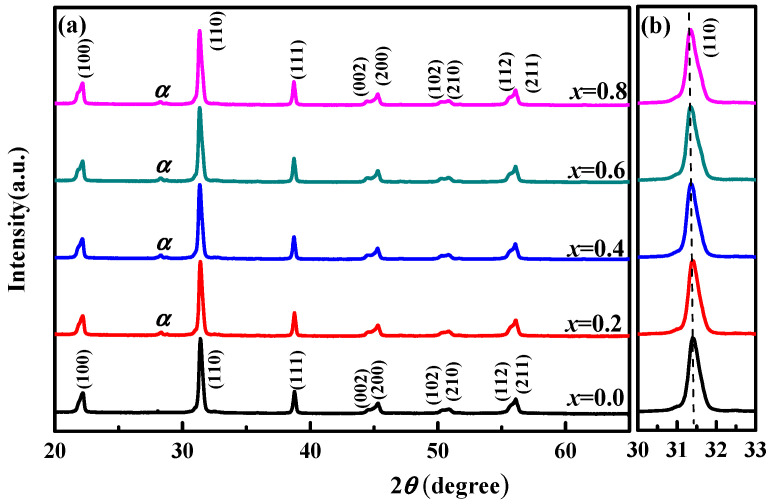
(**a**)X-ray diffraction pattern of BSI–PT–PMN–*x*Co Ceramics (Cobolt modified 0.40Bi(Sc_3/4_In_1/4_)O_3_–0.58PbTiO_3_–0.02Pb(Mg_1/3_Nb_2/3_)O_3_ ), (**b**) The variation trend of the selected peak (1 1 0). Note: α refers to Co_2_O_3_.

**Figure 2 micromachines-12-01556-f002:**
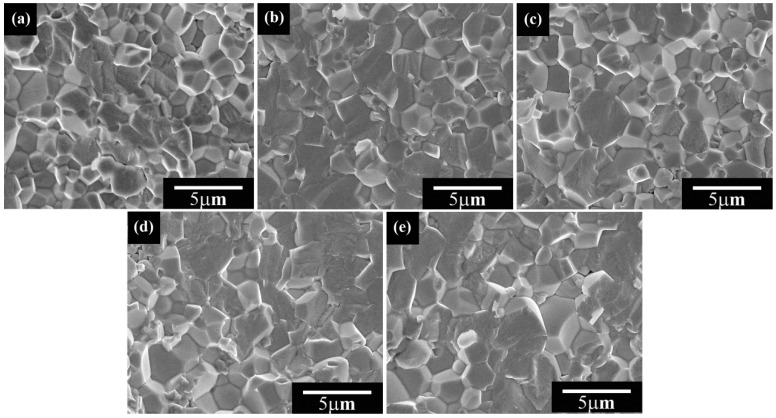
Microstructure of faces for BSI–PT–PMN–*x*Co, (**a**) *x* = 0, (**b**) *x* = 0.2, (**c**) *x* = 0.4, (**d**) *x* = 0.6, (**e**) *x* = 0.8.

**Figure 3 micromachines-12-01556-f003:**
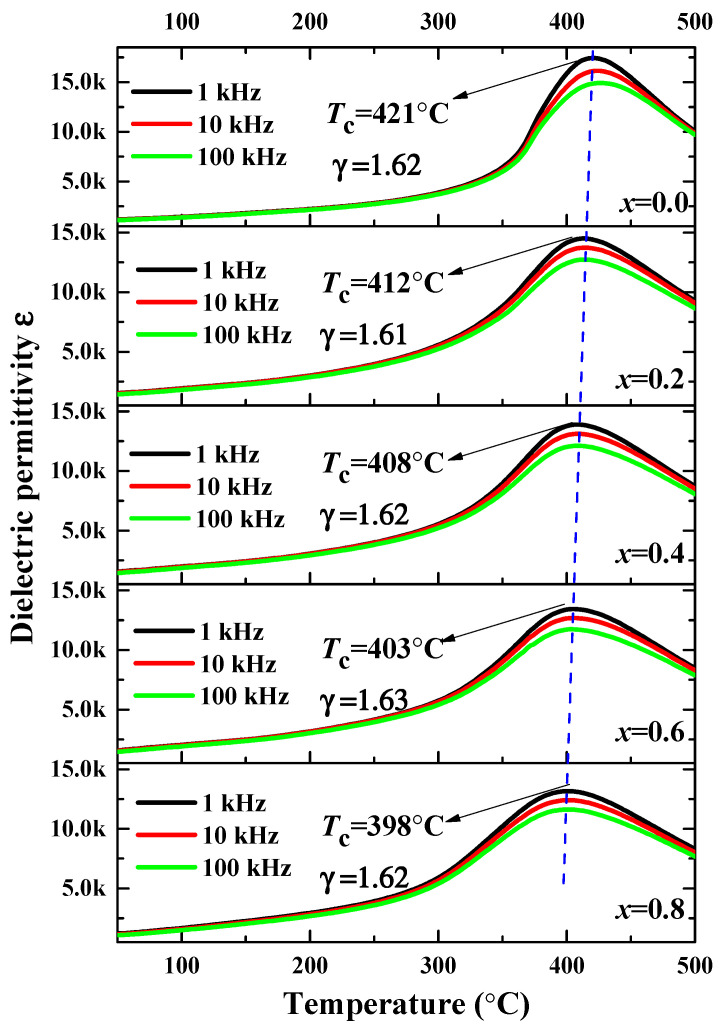
Dielectric permittivity (at 1 kHz, 10 kHz, and 100 kHz) as a function of temperature for BSI–PT–PMN–*x*Co (*x* = 0, 0.2, 0.4, 0.6, 0.8).

**Figure 4 micromachines-12-01556-f004:**
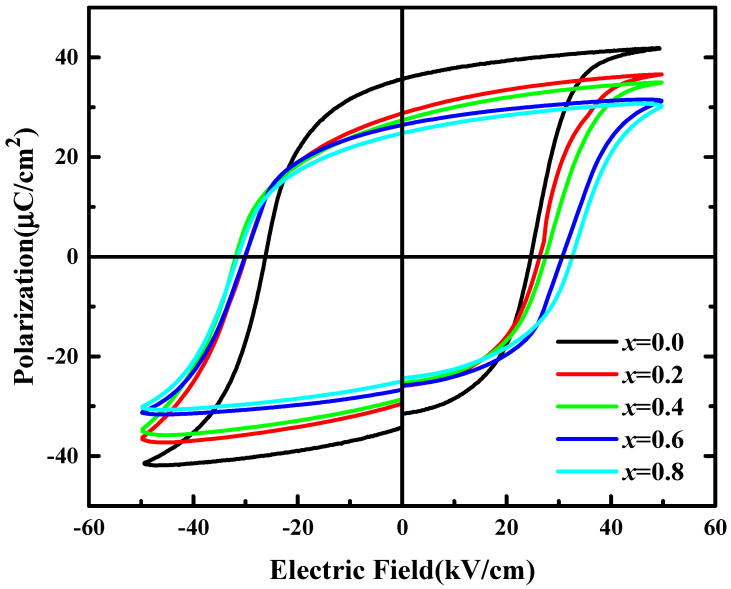
Bipolar field induced polarization for BSI–PT–PMN–*x*Co (*x* = 0, 0.2, 0.4, 0.6, 0.8).

**Figure 5 micromachines-12-01556-f005:**
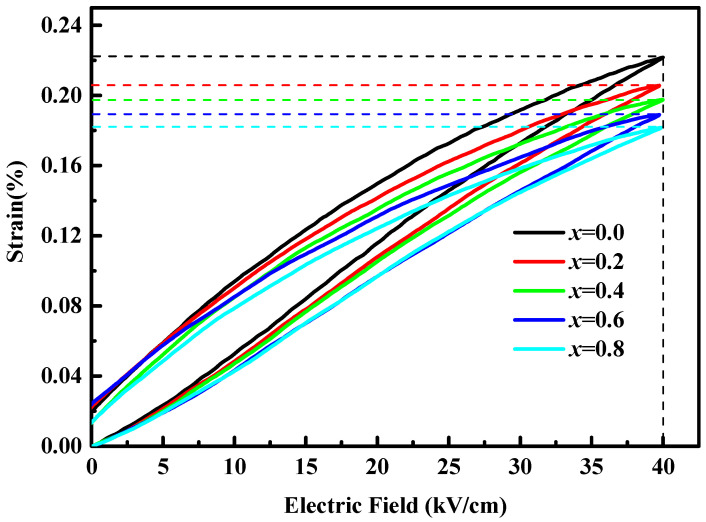
Unipolar field induced strain for BSI–PT–PMN–*x*Co (*x* = 0, 0.2, 0.4, 0.6, 0.8).

**Figure 6 micromachines-12-01556-f006:**
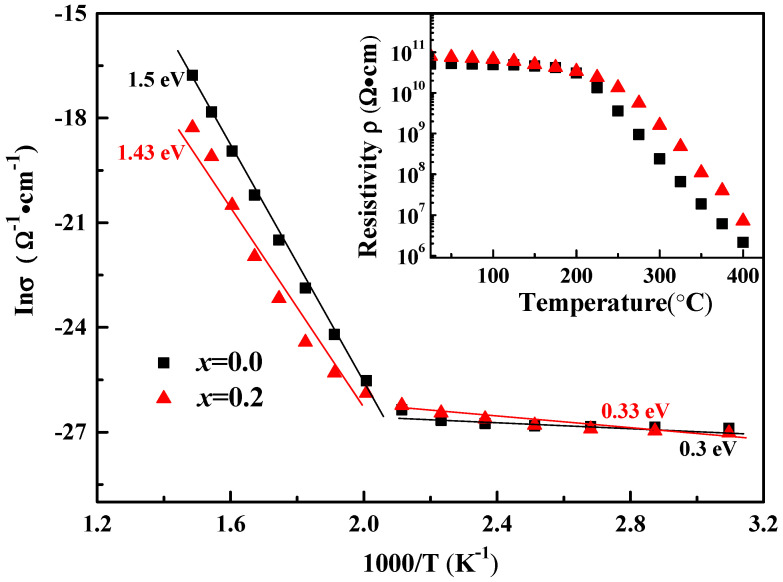
Arrhenius-type plots of conductivity *σ* as a function of temperature for BSI–PT–PMN with undoped and 0.2% doped Co ion. Inset depicting the resistivity *ρ* as a function of temperature.

**Figure 7 micromachines-12-01556-f007:**
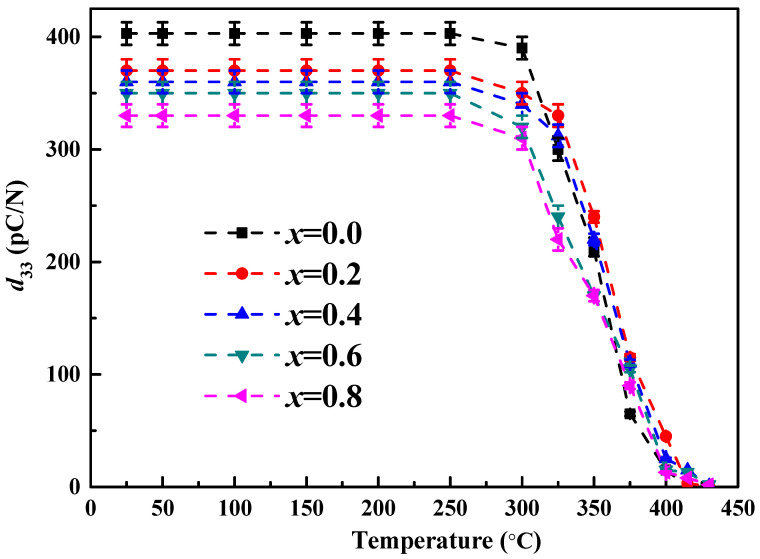
Piezoelectric constant *d*_33_ as a function of temperature for BSI–PT–PMN–*x*Co (*x* = 0, 0.2, 0.4, 0.6, 0.8).

**Table 1 micromachines-12-01556-t001:** Structure parameter of BSI–PT–PMN–*x*Co ceramics.

Material	*a* *(*Å*)*	*c* *(*Å*)*	*c/a*	Relative Density (%)	Grain Size (μm)
*x* = 0.0	3.920	4.005	1.022	96.2	1.94
*x* = 0.2	3.921	4.004	1.021	96.3	2.32
*x* = 0.4	3.922	4.003	1.020	97.1	2.41
*x* = 0.6	3.922	3.998	1.019	95.3	2.53
*x* = 0.8	3.921	3.989	1.017	95.6	2.68

**Table 2 micromachines-12-01556-t002:** Room temperature electrical properties of Co ion-doped BSI–PT–PMN ceramics.

Material	*T*_c_ (°C)	*ε* _r_	tan*δ*	*d*_33_ (pC/N)	*k_p_* (%)	*E_c_* (kV/cm)	*P_r_* (μC/cm^2^)	Strain (%)	*d*_33_* (pm/V)	*h* (%)
*x* = 0.0	421	1685	0.055	403	47.2	24.7	36.4	0.222	555	17.8
*x* = 0.2	412	1450	0.051	370	46.5	27.1	29.2	0.205	512.5	16.0
*x* = 0.4	408	1433	0.048	360	45.7	28.6	28.4	0.198	495	14.5
*x* = 0.6	403	1386	0.043	350	43.8	30.4	26.3	0.188	470	15.4
*x* = 0.8	398	1331	0.044	330	41.6	32.3	25.1	0.182	455	14.6

## Data Availability

The data used to support the findings of this study are available from the corresponding author upon request.

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
