# Peer review of "Preparation, Structure, and Electrical Properties of Cobalt-Modified Bi(Sc3/4In1/4)O3–PbTiO3–Pb(Mg1/3Nb2/3)O3 High-Temperature Piezoelectric Ceramics"

_micromachines, 2021, doi:10.3390/mi12121556_

Round 1

Reviewer 1 Report

  1. The title should be changed, it is confusing, "BSI-PT-PMN" unclear to readers.
  2. The abstract should be written properly.
  3. Co2+ or Co3+ ion?
  4. The introduction part needs to be written in context to current developments in lead-based materials and context to Co as well as other acceptor dopants.
  5. The introduction part is very shot and doesn't reflect the motivation and scope of this work.
  6. Is Tc of PZT is low? Since authors claimed drawbacks like low Tc.
  7. The scientific explanation for observed results is now well-written. Should be written clearly and emphasized on the pinpoint effect of Co ions substitution on the properties of 40Bi(Sc3/4In1/4)O3-0.58PbTiO3-0.02Pb(Mg1/3Nb2/3)O3 ceramics.
  8. The effective piezoelectric constant i.e. d33* values are missing.
  9. The authors should discuss in detail, how the measurement of d33 as a function of temperature was taken.
  10. In the introduction author state, Co improves high-temperature resistivity and maintains the Tc, explain the proper reason behind it?
  11. How does the author confirm the pure perovskite phase, which method or any database was used to confirm the pure phase? What is the crystal structure of BSI-PT-PMN-xCo?
  12. Near to 2θ = 28°, there is a small peak that appears which is not indexed, then how Does the author say there is no secondary phase formation?
  13. Why does Co ion facilities the grain growth of BSI-PT-PMN-xCo?
  14. With an increase in Co content, the T c decreases with an increase in broadening of Tc peak, Explain?
  15. Explain in detail the relation between oxygen vacancies and domain wall pinning corresponding to the P-E hysteresis loop?
  16. Figure 7 clearly shows the stabilization of the piezoelectric constant up to the temperature 325 °C for all compositions but the piezoelectric constant d33 decreases with an increase in the Co content, then how is the Co is beneficial for actuator application in terms of piezoelectric constant?

Reviewer 2 Report

Manuscript ID 1475051.

Article Type: Article

Title: Research of Co Ion Doped BSI-PT-PMN Ceramics for High Temperature Actuator Applications
Journal: Micromachines

Referee Review

The authors report on the results of preparation and complex investigation of the 0.40Bi(Sc3/4In1/4)O3-0.58PbTiO3-0.02Pb(Mg1/3Nb2/3)O3 ceramics doped with different Co ion amount (x=0, 0.2, 0.4, 0.6, 0.8 wt.%). Such ceramics are of great practical interest for use in piezoelectric devices, in particular, in actuators. In this regard, I believe that the topic of the article corresponds to the Journal Micromachines. Nevertheless, I disagree with one of the main statements of the article that BSI-PT-PMN-0.2Co promising candidate for high-temperature actuator applications. In fact, Co-modification leads to the decline of all key functional parameters for high-temperature actuator applications (remark 5). I suggest accepting this manuscript after minor revision (mainly corrections to minor methodological errors and text editing). Next, I brought criticism.

  1. In the Introduction section Authors note that “composition modification [16-22] and introducing a third component [23-27] are two effective and widely accepted methods to improve the comprehensive performance of piezoelectric ceramics.” But in fact, introducing a third component is rarely results in an increase in both the piezoelectric response and the Curie temperature relative to the binary BS-PT system. For example systems with the third component PbNi1/3Nb2/3O3 [Jpn. J. Appl. Phys. 52(10R), 101101(2013). DOI: 10.7567/JJAP.52.101101], PbMg1/3Nb2/3O3 [J. Am. Ceram. Soc. 101(2), 683 (2018). DOI: 10.1111/jace.15225], PbZn1/3Nb2/3O3 [J. Appl. Phys. 109(1), 014105 (2011). DOI: 10.1063/1.3525995], Bi(Zn1/2Ti1/2)O [ Mater. Chem. C 6, 456 (2018). DOI: 10.1039/c7tc04975g], Pb(Cd1/3Nb2/3)O3 [J. Appl. Phys. 126, 234103 (2019). DOI: 10.1063/1.5126065] and Pb(Sb1/2Nb1/2)O3 [J. All. Compd. 731, 1140 (2018). DOI: 10.1016/j.jallcom.2017.10.052 ] demonstrate higher values of d33 but with lower Curie temperatures (Tc). On the other side, addition the third component BiGaO3 [J. Am. Ceram. Soc. 91(9), 2943 (2008). DOI: 10.1111/j.1551-2916.2008.02580.x] leaves to increase of the Curie temperature with the decrease of d33. But simultaneously increasing both d33 and Tc is a challenge when creating materials based on the binary BS-PT system. I think that a small overview indicating this fact should be added to the Introduction section.
  2. In the Results and Discussion section, the values of the relative density and average grain size of ceramics with different cobalt concentrations should be given.
  3. For temperature dependences of the dielectric permittivity, the estimation of the peak diffuseness (diffuseness parameters δ and γ) is needed [see details, for example in Acta Materialia 193 (2020) 40-50. DOI: 1016/j.actamat.2020.04.035; or Phys. Rev. B. 98 (2018) 174104. DOI: 10.1103/PhysRevB.98.174104].
  4. For strain behaviors (Fig. 5) the estimation of the large-signal d33 (S/E) is required.
  5. My main criticism is related to the statement that “BSI-PT-PMN-0.2Co promising candidates for high-temperature actuator applications”. In fact, Co-modification leads to the decline of all key functional parameters for high-temperature actuator applications: electric field-induced strain (Fig. 5), temperature stability (Fig. 7), d33 (Table 1). Resistivity is not a critical parameter for actuator applications. This is why, in my opinion, the title of the article and the statement that increasing the concentration of cobalt favors use in high-temperature actuators is inappropriate.

Reviewer 3 Report

Page 2 In the last of introduction, the author said that this work aims to generate good electrical performance but still provides a relatively high Curie temperature. But with Co ion doping, Pr, Tc and d33 are all degraded. What is the main purpose of this work? Or what are the advantages by doping Co ions?

Page 3 It is mentioned that the Co3+ and Co2+ are coexistent for Co2O3 used as dopants in this work. Do Co3+ and Co2+ have the same effect on electrical properties?

Page 4 Figure 3 shows that with the increase in Co ion doped content, Curie temperature slightly decreases. The authors explained that this is attributed to the slight lattice distortion caused by B-site substitution. However, smaller Co ion substitution can induce lattice contraction to lower the symmetry of the phase structure and the lowered symmetry can enhance the resistance of the ceramics to phase transformation when temperature improves. As a result, Curie temperature should be increased. It is better to check this description.

Page 4 The doping of Co ions can facilitate the grain growth and the grain sizes of BSI-PT-PMN-xCo ceramics become slightly larger when Co ion content increased. How does grain size affect the electrical properties?
